# Nurse-led psychoeducational interventions in patients suffering from schizophrenia or other psychotic disorders and their families: A scoping review protocol

Carla Gramaglia[1¤a☉], Eleonora Gambaro[1☉], Lucia Bestagini[2], Erika Bassi[2], Alberto Dal Molin[2‡], Patrizia Zeppegno[1¤a‡]*

**1** S.C. Psichiatria, Azienda Ospedaliero Universitaria Maggiore della Carità, Novara, Italy, **2** Department of Translational Medicine, University of Piemonte Orientale, Novara, Italy

☉ These authors contributed equally to this work.
‡ These authors also contributed equally to this work.
¤a Current Address: Institute of Psychiatry, Department of Translational Medicine, University of Piemonte Orientale, Novara, Italy
* patrizia.zeppegno@med.uniupo.it

## Abstract

### Background

Mental health conditions are one of the most significant public health issues. Psychoeducation is an organized, structured, and understanding transfer of knowledge about the disease and its treatment, incorporating inspirational and educational elements to allow patients to achieve better management of their disease and increased treatment adherence and effectiveness, especially for patients with schizophrenia. This review will consider studies that include adults suffering from schizophrenia or other psychotic disorders and their family members. In this scoping review, the rigor expected of all primary and secondary research will be guided by clear and consistent standards governing its design, regardless of the level of inquiry. The review will aims to map the currently available research in nurse-led psychoeducational programs conducted on these populations.

### Inclusion criteria

Adults suffering from schizophrenia or other psychotic disorders and their family members; patients and family members aged from 18 to 75 years old. Both experimental and quasi-experimental study designs, including randomised controlled trials, non-randomised controlled trials, before-and-after studies, interrupted time-series studies, analytical observational studies, and descriptive observational study designs. Intervention: a) psychoeducational interventions on patients suffering from

**Data availability statement:** No datasets were generated or analysed during the current study. All relevant data from this study will be made available upon study completion

**Funding:** The author(s) received no specific funding for this work.

**Competing interests:** The authors have declared that no competing interests exist.

**Abbreviations:** DSM-5-TR, Diagnostic and Statistical Manual of Mental Disorders, Fifth Edition, Text Revision; JBI, Joanna Briggs Institute methodology for scoping reviews, PRISMA-ScR, Preferred Reporting Items for Systematic Reviews and Meta-Analyses extension for Scoping Reviews.

schizophrenia and/or other psychotic disorders and their families, b) nurse-led psychoeducational interventions.

## Methods and analysis

The scoping review will be conducted following the Joanna Briggs Institute methodology for scoping reviews. Medline, Scopus and CINAHL databases will be searched for relevant studies. The reference list of the selected articles will be screened. Data will be summarised in a comprehensive narrative summary to highlight the following for each nurse-led intervention program: main features, setting, and target.

## Ethics and dissemination

This study does not require ethical approval. The study will be submitted to a peer reviewed journal, will be publicly disseminated and will be the topic of research presentations.

## Review registration number

Registered in the Open Science Framework on April 6, 2023. OSF Registration Doi: https://doi.org/10.17605/OSF.IO/ASQ95.

### Strengths and limitations

1. Focus on the importance of nursing care in discussing the patient with schizophrenia and the family, on the needs they have nurses take on a more active role in the treatment and recovery process.

2. We will carry out this scoping review because the presence of the aforementioned elements has not been documented in extant studies of a similar nature. Therefore, it is believed that it can contribute to the development of research and in-depth studies, to outline an effective and reproducible intervention in inpatient and outpatient settings on patients with schizophrenia and their caregivers.

3. The strengths of this review are the use of the PRISMA methodology for selection and the synthesis of the articles, and the use of three different databases (Scopus, PubMed and Cinahl). People with experience in the field, such as nurses or psychiatrists, will mainly be responsible for study selection and data extraction.

4. However, although our review suggests the positive aspects of nurses' involvement in psychoeducational interventions for the treatment of patients with schizophrenia and their families, the generalizability of the results may be limited. In fact, due to the nature of scoping review, a formal quality assessment of the included studies will not be conducted. However, a brief comment

on the methodological rigour of the included studies will be provided, noting any limitations or biases that may have influenced the interpretation of the results.

5. In the first place, we will not perform a formal risk of bias assessment for each study, only articles in English will be selected, and there will be no consultation of the grey literature. Scoping reviews sometimes exclude grey literature due to its potential lower quality or lack of peer review. However, this can lead to missing valuable unpublished or non-commercial research, particularly in fields where nurse-led interventions may not be widely published in peer-reviewed journals. Limiting grey literature searches in this scoping review will require balancing the need for comprehensiveness with the available time and resources for searching and evaluating non-standard sources. Second, most of the studies were from high-risk countries income and their applicability across different contexts may be limited.

## Introduction

Mental health disorders are among the greatest public health concerns, causing both extreme human suffering and substantial financial loss [1]. According to the Diagnostic and Statistical Manual of Mental Disorders, Fifth Edition, Text Revision (DSM-5-TR), psychotic disorders include schizophrenia, other psychotic disorders, and schizotypal personality disorders. Five symptom domains define this spectrum: delusions, hallucinations, disorganized thinking, disorganized or abnormal motor behaviour (including catatonia) and negative symptoms, mainly represented by reduced emotional expression and avolition [2]. Negative symptoms may also manifest as difficulty in maintaining hygiene and satisfying self-care [3].

Schizophrenia or other psychotic disorders are highly debilitating chronic conditions, resulting in poor social functioning and low quality of life [3]. Particularly, schizophrenia is a persistent, extreme, and disabling disease which affects approximately 1% of the population worldwide, affecting all cultures and all socio-economic groups [4]. The long-lasting and severe symptoms of schizophrenia cause considerable impairment and special needs. It has been proposed that schizophrenia is an illness that can be worsened by environmental stressors such as life events [5]. Individuals with schizophrenia might have deficits in problem-solving abilities. This might impact social and independent functioning and hinder the quality of life [6,7]. Insufficient coping abilities typically precede the full-blown onset of the disease however, medication often has little result on behavioural deficits [7]. As a result, there have been programs developed to improve problem-solving skills in schizophrenia, such as problem-solving therapy. Problem-solving therapy is a quick, focused type of psychiatric therapy. The treatment includes a couple of practical sessions focused on the patient's current problems, identifying tasks and methods of solving them [8].

Those with severe mental illness, specifically schizophrenia or other psychotic disorders, typically have little to no insight regarding the existence of their health problem. Whether this lack of insight is primary or related to a denial of the condition, both situations ultimately increase non-compliance [9].

For all these reasons, psychoeducational interventions become fundamental in assisting the pharmacological treatment of patients with schizophrenia or other psychotic disorders. Psychoeducation offers a systematic and structured transfer of knowledge about the illness and its treatment, integrating emotional and motivational aspects to enable patients to cope with the illness and to improve their treatment adherence and efficacy [9]. It is not simply about 'providing information', but rather about enhancing patient education aimed at promoting awareness, providing tools to manage, cope with and live with a chronic psychiatric condition and changing behaviours and attitudes related to the condition [10]. Furthermore, psychoeducation aims at supporting family members' understanding of what it means to live with an illness, to enhance their ability to assist the patient and to support treatment providers in the treatment program [11].

A vast body of literature has pointed to the major role of psychoeducation in supporting patients affected by schizophrenia or other psychotic disorders and their family members. A systematic review of the "effectiveness of family interventions

on psychological distress and expressed emotion in family members of individuals diagnosed with first-episode psychosis" [3] underscored the possible impact of psychosis on patients' families, both from an emotional and practical point of view. Caregivers may feel overwhelmed by approaching mental health services and dealing with hospitalization and stigma. [4] Data synthesis of the three individual studies suggested that there was no available evidence to examine the effectiveness of family interventions on family caregivers' psychological distress and expressed emotion during the onset of first-episode psychosis (FEP). Another review [3] highlights the critical role of family interventions for individuals experiencing first-episode psychosis (FEP). The study emphasises the effectiveness of various interventions, including education, psychoeducation, communication skills training, and cognitive-behavioural therapy, in significantly reducing psychological distress among family members. The review stresses the importance of commencing treatment within two years of FEP onset, asserting that timely intervention can yield more favourable outcomes, including remission.

There were no statistically significant interventions that addressed psychological distress and expressed emotion in family members who live with and care for persons with first-episode psychosis. There is insufficient evidence available to evaluate the effect sizes for pooled outcomes.

Nurses play a core role in patients' psychoeducation [12], even though their role is still perceived as ambiguous, especially in outpatient settings. The establishment of advanced practice nurses, formerly known as nurse specialists in the field of psychiatric care, has recently tried to meet this service need and has the goal of preventing hospitalization and/or relapse in patients with first-onset mental illness. Psychiatric advanced practice nurses are employed to address the psychiatric symptoms of patients with mental health problems, seeking high-quality and timely care for those health problems and needs, as expected by mental health care services and consumers [13,14].

Nurses' role can and should be pivotal in psychiatric health services [15], also considering the evolving challenges and needs entailed by psychiatric patients' treatment, both in the inpatient and outpatient facilities [16]. There are many interventions performed by nurses in the adult psychiatric outpatient setting to help patients and family members work on their strengths and improve coping [12]. Nonetheless, their function has been referred to as hard to clarify, [17] blurred [18] and uncertain [19].

A better understanding of the factors supporting, and, on the other hand, hindering the role of nurses in providing evidence-based treatments is needed. The effectiveness of psychoeducational therapies administered by nurses with advanced psychiatric training for newly identified patients referred to psychiatric care (particularly patients with schizophrenia or other psychotic disorders) has been demonstrated by a relatively small number of controlled trials and, consequently, based on limited clinical data. Moreover, there are no studies that analyze the nurse-led psychoeducation interventions carried out exclusively on both patients with schizophrenia and their families.

More in detail, a preliminary search of PROSPERO, MEDLINE, the Cochrane Database of Systematic Reviews, and JBI Evidence Synthesis was conducted, and no current or in-progress scoping reviews or systematic reviews on the topic were identified. Therefore, this scoping review is the first that aims to address this knowledge gap, mapping the currently available research in nurse-led psychoeducational programs conducted on patients suffering from schizophrenia or other psychotic disorders and their families.

## Research question(s)

To accomplish the aim of the study, the following research question was identified

*Which nurse-led psychoeducation programs have been implemented for outpatient or inpatients suffering from* schizophrenia or other psychotic disorders *and their families?*

To answer this question the following sub-questions were identified:

1. What are the characteristics of nurse-led intervention programs? The characteristics to be extracted for are the name of the program; the objective of the intervention; frequency of sessions; type of intervention; intervention facilitators; evaluation; and implementation context.

2. In what settings (outpatient, inpatient, community) are the nurse-led programs implemented and evaluated?

3. Who is the target of the nurse-led intervention programs? The characteristics of the populations to be described are age, gender, level of education, marital status, biological relationship with the patient, monthly family income, number of family members living with the patient, other family members with psychiatric illness, diagnosis, duration of illness, current medication and psychiatric treatment they are receiving, previous hospitalization and duration of hospitalization in a psychiatric setting, adherence to treatment, disease insights, intervention frequency.

4. Do the nurses have a specific advanced degree?

## Eligibility criteria

### Participants

Adults, aged over 18 and under 75, suffering from schizophrenia or other psychotic disorders and their family members, aged over 18 and under 75, too.

### Concept

The concept to be studied is nurse-led psychoeducational implemented in outpatient or inpatient settings for psychiatric patients with schizophrenia or other psychotic disorders and their families. The term psychoeducation refers to a type of intervention offering systematic and structured knowledge transfer about the illness and its treatment, integrating emotional and motivational aspects to enable patients' coping with the illness and improve their treatment adherence and efficacy. Psychoeducational programs provide both disease-specific (e.g., early recognition and management of relapse symptoms, implications of the illness) and general information (e.g., promotion of healthy lifestyle, problem-solving and communication skills training, identification of stressors in the household), and education of family members and primary caretakers. Furthermore, psychoeducation aims at supporting family members' understanding of what it means to live with an illness, to enhance their ability to assist the patient and to support treatment providers in the treatment program. Specifically, this scoping review focuses on nurse-led interventions aimed at providing care to the person with schizophrenia or other psychotic disorders and family caregivers. A nurse-led psychoeducational intervention is a healthcare approach in which registered nurses assume the primary role in the design, implementation, and supervision of various aspects of patient care and treatment. The proposed scoping review will encompass studies that are predicated on the competencies, knowledge, and skills inherent to the nursing profession. Nurse-led interventions are implemented in a variety of healthcare settings, including hospitals, clinics, community health centers, and even in patients' homes. Nurse-led self-management programs have emerged as a promising strategy that emphasizes patient education, empowerment, and active involvement in patient care. Nurse-led programs have precipitated a paradigm shift, thereby empowering patients to assume an active role in their health and well-being [20].

Research that is exclusively focused on standard interventions and/or counseling provided by mental health services will be excluded from consideration.

### Context

Any type of outpatient or inpatient setting of psychiatric care; intervention implemented in any setting (hospital, outpatient clinic, mental health centre, etc.).

### Types of sources

This scoping review will consider both experimental and quasi-experimental study designs including randomized controlled trials, non-randomized controlled trials, before and after studies and interrupted time-series studies. In

addition, analytical observational studies including prospective and retrospective cohort studies, case-control studies and analytical cross-sectional studies will be considered for inclusion. This review will also consider descriptive observational study designs including case series, individual case reports and descriptive cross-sectional studies for inclusion.

Qualitative studies will also be considered that focus on qualitative data, including, but not limited to, research methods/methodologies and paradigms such as phenomenology, grounded theory, ethnography, qualitative description, action research and feminist research.

## Methods

The proposed scoping review will follow the JBI methodology for scoping reviews [21], and in line with the Preferred Reporting Items for Systematic Reviews and Meta-Analyses extension for Scoping Reviews (PRISMA-ScR) [22]. This study does not require ethical approval.

The review protocol has been registered within the Open Science Framework database (Doi: https://doi.org/10.17605/OSF.IO/ASQ95).

### Search strategy

An initial search limited to PubMed was undertaken to identify articles on the topic. The text words contained in the titles and abstracts of relevant articles and the index terms used to describe the articles were used to develop a full search strategy for Medline, Scopus and CINAHL (Supp 1). The search strategy, including all identified keywords and index terms, will be adapted for each included database and/or information source. The reference list of all included sources of evidence will be screened for additional studies. Peer-reviewed journal articles written in English and Italian will be included.

### Study selection

The literature results were exported from the databases to the Covidence reference management system (https://www.covidence.org/). Duplication of search results across the databases was eliminated using Zotero reference management software. Before the analysis began, all screeners received training on how to use the Covidence system and on the content area (i.e., schizophrenia and nurse-led psychoeducative intervention). Titles and abstracts will then be screened by two independent reviewers according to the inclusion criteria for the review. Titles and abstracts written in languages other than English or Italian will be excluded, as well as duplicate articles. The full text will be assessed by independent reviewers. Reasons for the exclusion of full texts will be recorded and reported. Records that refer to the same study will all be read but will beconsidered unique. Any disagreement arising between the reviewers at each stage of the selection process will be resolved through discussion, and/or with an additional reviewer. Furthermore, the snowballing method will be employed [23]. Specifically, studies that meet the inclusion criteria derived from the screening of the reference lists of reviewed articles will be included to allow for the incorporation of missing articles deemed necessary in this scoping review to identify further correlated evidence.

The results of the search and the study inclusion process will be reported in full in the final scoping review and presented in a PRISMA flow diagram. [7] (Supp 2).

The screening phase will be completed by July 2025

### Data extraction

Data will be extracted from studies included in the scoping review by two independent reviewers using a data extraction tool developed by the reviewers. The data extracted will include specific details about the participants, concept, context,

study methods and key findings relevant to the review questions. The following data categories will be extracted for each article, when applicable:

- Authors, study location, publication details

- Study design

- Aim of the research

- Setting

- Participants

- Data source, data analysis

- Psychoeducational intervention

- Factors affecting the psychoeducational intervention

- Healthcare professionals conducting the intervention

- Specifically advanced degree nurses

- Outcome of the nurse-led program

All results will undergo double data entry. For dichotomous data, the number of cases (n) and the total sample size (N) will be extracted. For continuous data, the mean, standard deviation (SD) and sample size will be extracted. If some of the included studies are unclear in terms of population, intervention or results, one of the reviewers will contact the author(s) to clarify the data.

This list will be modified and revised as necessary during the process of extracting data from each included evidence source. Each time a change is made, the list will be revised. When appropriate, authors of studies will be contacted to request missing data. The scoping review will extract the results and map them descriptively (rather than analytically). Thus, we will not attempt to assess the certainty of these findings or summarise them as one would in a systematic review.

The entire scoping review process will be completed at the end of September 2025.

## Data analysis and presentation

The data will be presented in tabular form. A narrative summary will accompany the tabulated results and will describe how the results relate to the objective and questions of the review, summarizing the extracted data around the key concepts and identified gaps in knowledge. As a scoping review can be used to map the concepts underpinning a research area, the findings from the scoping review will provide an overview of the research rather than an assessment of the quality of the individual studies. The scoping review will provide an in-depth overview of the field, avoiding a simple summary of the evidence regarding the potential impact of different psychoeducational interventions. In addition, the scoping review will focus on identifying gaps in the current literature rather than generating new knowledge. In line with typical scoping reviews, we will not assess the quality or bias of studies, nor will we conduct a systematic assessment of the external validity of evidence, such as a GRADE rating. Rather, we will outline the main characteristics of the most reliable evidence in this area and provide comments on its relevance in various contexts.

## Ethics and dissemination

The study will be submitted to a peer-reviewed journal, will be publicly disseminated and will be the topic of research presentations. The approach to conducting and reporting the scoping review will be congruent with the PRISMA-ScR checklist. The scoping reviews will be registered with the Open Science Framework (https://osf.io/).

## Supporting information

**S1 Supplemental material 1:   Search strategy.**
(DOCX)

**S2 Supplemental material 2:   PRISMA 2020 flow diagram for scoping reviews which included searches of databases and registers only.**
(TIF)

**S1 File.  Author Formatting Checklist.**
(DOCX)

## Acknowledgments

This research was conducted in the context of the Aging Project – Department of Excellence,
Università del Piemonte Orientale, Novara, Italy. The completion of this research project would not have been possible without the contributions and support of all authors. We are deeply grateful to all those who played a role in the success of this project.

## Author contributions

**Conceptualization:** Carla Gramaglia, eleonora gambaro, Erika Bassi, Alberto Dal Molin.

**Methodology:** Carla Gramaglia, Erika Bassi, Alberto Dal Molin.

**Supervision:** Alberto Dal Molin, Patrizia Zeppegno.

**Validation:** Patrizia Zeppegno.

**Visualization:** Patrizia Zeppegno.

**Writing – original draft:** eleonora gambaro, Lucia Bestagini.

**Writing – review & editing:** Carla Gramaglia, eleonora gambaro, Erika Bassi, Alberto Dal Molin, Patrizia Zeppegno.

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
