## [Decision Letter · Decision Letter 0]

Dear Dr. gambaro,

Thank you for submitting your manuscript to PLOS ONE. After careful consideration, we feel that it has merit but does not fully meet PLOS ONE’s publication criteria as it currently stands. Therefore, we invite you to submit a revised version of the manuscript that addresses the points raised during the review process.

**A major revision of the method section is necessary with more details on all key aspects (research questions, inclusion/exclusion criteria, what is a study of a particular interest, etc.) This section gives the impression of a scoping rather than a systematic review therefore please clarify all necessary points as stated by the Reviewers.**

We look forward to receiving your revised manuscript.

Kind regards,

Eleni Petkari

Academic Editor

PLOS ONE

**Journal Requirements:**

1. When submitting your revision, we need you to address these additional requirements. Please ensure that your manuscript meets PLOS ONE's style requirements, including those for file naming. The PLOS ONE style templates can be found at https://journals.plos.org/plosone/s/file?id=wjVg/PLOSOne_formatting_sample_main_body.pdf and https://journals.plos.org/plosone/s/file?id=ba62/PLOSOne_formatting_sample_title_authors_affiliations.pdf 2. We noticed you have some minor occurrence of overlapping text with the following previous publication(s), which needs to be addressed: -DOI: 10.11124/jbisrir-2014-662 In your revision ensure you cite all your sources (including your own works), and quote or rephrase any duplicated text outside the methods section. Further consideration is dependent on these concerns being addressed. 3. Please amend either the abstract on the online submission form (via Edit Submission) or the abstract in the manuscript so that they are identical. 4. Please include captions for your Supporting Information files at the end of your manuscript, and update any in-text citations to match accordingly. Please see our Supporting Information guidelines for more information: http://journals.plos.org/plosone/s/supporting-information.

Reviewers' comments:

Reviewer's Responses to Questions

**Comments to the Author**

1. Does the manuscript provide a valid rationale for the proposed study, with clearly identified and justified research questions?

Reviewer #1: Partly

Reviewer #2: Yes

2. Is the protocol technically sound and planned in a manner that will lead to a meaningful outcome and allow testing the stated hypotheses?

Reviewer #1: Yes

Reviewer #2: Yes

3. Is the methodology feasible and described in sufficient detail to allow the work to be replicable?

Reviewer #1: Yes

Reviewer #2: No

4. Have the authors described where all data underlying the findings will be made available when the study is complete?

Reviewer #1: No

Reviewer #2: Yes

5. Is the manuscript presented in an intelligible fashion and written in standard English?

Reviewer #1: Yes

Reviewer #2: No

You may also provide optional suggestions and comments to authors that they might find helpful in planning their study.

**Reviewer #1: ** PLOS ONE REVIEW

The manuscript (ms) presents a scoping review protocol aimed at mapping existing research on nurse-led psychoeducational interventions for patients with schizophrenia or other psychotic disorders and their families. This is a relevant and important topic, as it highlights the potential role of nurses in delivering structured psychoeducational programs that may improve treatment adherence and patient outcomes. However, while the study aims to address an important gap in the literature, several sections require further clarification and refinement before I can recommend the ms for publication.

While the authors emphasize the limited clinical evidence regarding nurse-led psychoeducational interventions, it remains unclear whether there is a sufficient foundation for conducting this review. Furthermore, if the effectiveness and outcomes of these interventions are not being assessed, the value of the review needs to be better articulated.

There are inconsistencies between the research questions and the manuscript’s content:

• The data analysis section states that an improvement in well-being and quality of life is expected, yet none of the research questions focus on assessing improvements.

• The manuscript states that it will extract information on whether interventions target (a) patients and their families or (b) are specifically nurse-led, but it is unclear which research question this aligns with. Furthermore, there is no clear plan for comparing these groups or specifying relevant outcome measures.

The study design is appropriate for addressing the research question, but more details on the inclusion and exclusion criteria for studies (beyond age range and study design) should be provided, particularly regarding the selection of interventions that qualify as "nurse-led psychoeducation.

• Systematic reviews will be included but the ms does not specify whether the review itself or only primary studies within those reviews will be considered.

• The inclusion of “studies of particular interest and meeting the inclusion criteria deriving from the screening of the reference lists of the reviewed articles” is vague. The authors should define what constitutes a study of “particular interest.”

While the manuscript outlines a structured approach to data extraction, several key elements remain unclear:

• There is no mention of how extracted data will be validated or cross-checked.

• “Articles discussing data from the same study will all be read, but only the most relevant ones will eventually be included“. However, the procedure on duplicate data is not clear.

• The extraction list will be modified and revised as necessary during the process of extracting data, but it is not clear if they will go back and re-review each article every time a modification is made.

• The ms does not clarify how the impact of interventions on patient outcomes (e.g., adherence, QOL) will be assessed in the included studies. Details on how outcome variables will be analyzed or summarized should be included.

The manuscript is generally well-written, but certain sections are repetitive and could be streamlined for clarity. For example, the age range of participants is mentioned multiple times. Additionally, some terms should be revised to align with person-first language (e.g., “schizophrenic patients” should be changed to “patients with schizophrenia”).

Several linguistic and conceptual inconsistencies should also be addressed:

• .. conditions are amongst the biggest public health issues, triggering both severe human .. could be streamlined to "one of the most significant public health issues"

• "The success of the nurse-led psychosocial therapies created and delivered by of nurse-led psychoeducational therapies created and delivered by advanced psychiatric practice nurses." A simpler version could be: "The success of nurse-led psychoeducational therapies delivered by advanced psychiatric practice nurses...”

• One of the research questions is: “What are the characteristics of nurse-led intervention programs? The characteristics to be searched for are the name of the program”. It should be “Extracted” instead of “searched”.

• The headline of a section is “review question”, but it is “research question” throughout the text

• “Qualitative studies will also be considered that focus on qualitative data including, but not limited to, designs such as phenomenology, grounded theory, ethnography, qualitative description, action research and feminist research.” However, these are technically not designs, but research methods/methodologies and paradigms.

• They switch between calling it papers/studies throughout the ms

Several methodological aspects require further clarification:

• The ms states that,”The strengths of this review are the use of the PRISMA methodology for selection and the synthesis of the articles and the use of three different databases (Scopus, PubMed and Cinahl), primarily carried out by people with experiences in the sector”. It is not clear what it is that are primarily carried out by people with experiences in the sector.

• It is not clear why “Standard interventions and/or consultations provided by mental health services will be excluded”?

• While the ms states that data will be publicly disseminated, it does not specify the repository or how the data will be made accessible. Please specify the repository where the data will be stored (e.g., OSF or Dryad) and outline measures for validating extracted data to ensure reliability.

• The ms does not include a preliminary PRISMA flow diagram, which is a key component of scoping review protocols. A diagram outlining the planned study selection process should be added

While the conclusions appropriately address the knowledge gap in nurse-led psychoeducational interventions, certain claims are made prematurely without supporting evidence. Specifically, the assertion that these interventions improve patient outcomes and treatment adherence is made without data to substantiate it. Additionally, the manuscript overgeneralizes the expected impact of the review on clinical practice. The decision to exclude grey literature may result in the omission of relevant studies, particularly in a field where nurse-led interventions may not be widely published in peer-reviewed journals. The potential limitations of excluding grey literature should be acknowledged and discussed.

The ms covers an important and timely topic, but several methodological and conceptual issues need to be addressed before it can be recommended for publication. The authors should:

1. Clarify the rationale for conducting this review and ensure alignment between the research questions and study objectives.

2. Provide more detail on inclusion/exclusion criteria, study selection, and data extraction.

3. Address inconsistencies and redundancies in terminology and writing.

4. Strengthen the methodological framework by specifying data validation methods and including a PRISMA flow diagram.

5. Moderate claims about the effectiveness of interventions and discuss the limitations of excluding grey literature.

**Reviewer #2: ** This is the protocol for a scoping review that will be conducted on studies implementing nurse-led psychoeducational interventions for patients and families of patients with psychotic spectrum disorders. The protocol is of interest; below are some brief comments:

The manuscript requires revision by a native English speaker. Examples of corrections: “Primary and secondary studies pertaining to both quantitative and qualitative paradigm.” This sentence appears incomplete; “Adults suffering from s schizophrenia”; “and or” should likely be “and/or”; “they do not appear to be present to date similar studies in the literature” should be rephrased; “When appropriate, authors of papers will be contacted to request missing” is an incomplete sentence; “we aspect” should probably be “expect.”

This is a protocol for a study that has not yet been conducted, so ensure that sentences are in the correct tense (e.g., “We carried out this scoping review” should be “We will carry out”).

Points 4 and 5 under Strengths and Limitations seem to imply that the study has already been conducted.

The in-text reference numbers appear after the period of the sentence they refer to. Ensure this is the correct format.

“The success of the nurse-led psychosocial therapies created and delivered by of nurse-led psychoeducational therapies created and delivered by advanced psychiatric practice nurses for patients who were recently identified and referred to psychiatric care (especially for patients suffering from schizophrenia or other psychotic disorders), however, has been demonstrated by a relatively small number of controlled trials and, consequently, limited clinical data.” This sentence is hard to read and contains errors.

In the subquestions of the review questions, I would add patient diagnoses to point 3.

Since it is already March 2025, I would update the target dates for the various steps of the review.

To ensure the validity of results, more information regarding the data extraction tool developed by reviewers is necessary, particularly concerning its validation if it is an automated tool for parts of the process not supervised by humans.

The time frame on which the research will focus must be specified, as per PRISMA guidelines. Up to which month in 2025 will inclusion criteria for the search apply?

In the supplementary materials regarding search strings, at least one string lists a year limit up to 2023.

**Do you want your identity to be public for this peer review?** For information about this choice, including consent withdrawal, please see our Privacy Policy

Reviewer #1: **Yes: ** lars clemmensen

Reviewer #2: No

---

## [Author Response · Author response to Decision Letter 1]

30 May 2025

Reviewer #1: PLOS ONE REVIEW

The manuscript (ms) presents a scoping review protocol aimed at mapping existing research on nurse-led psychoeducational interventions for patients with schizophrenia or other psychotic disorders and their families. This is a relevant and important topic, as it highlights the potential role of nurses in delivering structured psychoeducational programs that may improve treatment adherence and patient outcomes. However, while the study aims to address an important gap in the literature, several sections require further clarification and refinement before I can recommend the ms for publication.

While the authors emphasize the limited clinical evidence regarding nurse-led psychoeducational interventions, it remains unclear whether there is a sufficient foundation for conducting this review. Furthermore, if the effectiveness and outcomes of these interventions are not being assessed, the value of the review needs to be better articulated.

There are inconsistencies between the research questions and the manuscript’s content:

• The data analysis section states that an improvement in well-being and quality of life is expected, yet none of the research questions focus on assessing improvements.

Sorry for the misunderstanding. We have underscored in the data analysis section that an improvement in well-being and quality of life will not be assessed in this scoping review, as this question would be more suitable for a systematic review

• The manuscript states that it will extract information on whether interventions target (a) patients and their families or (b) are specifically nurse-led, but it is unclear which research question this aligns with. Furthermore, there is no clear plan for comparing these groups or specifying relevant outcome measures.

Sorry for the misunderstanding. We have underscored in the data analysis section that an improvement in well-being and quality of life will not be assessed in this scoping review, as this question would be more suitable for a systematic review

The study design is appropriate for addressing the research question, but more details on the inclusion and exclusion criteria for studies (beyond age range and study design) should be provided, particularly regarding the selection of interventions that qualify as "nurse-led psychoeducation.

We added a section to the inclusion criteria regarding the selection process for studies on nurse-led psychoeducational interventions.

• Systematic reviews will be included but the ms does not specify whether the review itself or only primary studies within those reviews will be considered.

Sorry for the misunderstanding. We have underscored in the data analysis section that an improvement in well-being and quality of life will not be assessed in this scoping review, as this question would be more suitable for a systematic review

• The inclusion of “studies of particular interest and meeting the inclusion criteria deriving from the screening of the reference lists of the reviewed articles” is vague. The authors should define what constitutes a study of “particular interest.”

In light of the provided recommendation, we made a revision of the sentence.

While the manuscript outlines a structured approach to data extraction, several key elements remain unclear:

• There is no mention of how extracted data will be validated or cross-checked.

We added a sentence as you suggested.

• “Articles discussing data from the same study will all be read, but only the most relevant ones will eventually be included“. However, the procedure on duplicate data is not clear.

In response to your suggestion, we have rephrased the sentence to provide a more thorough elaboration on the concept.

• The extraction list will be modified and revised as necessary during the process of extracting data, but it is not clear if they will go back and re-review each article every time a modification is made.

Thank you for your suggestion. We have added a more detailed explanation to the text.

• The ms does not clarify how the impact of interventions on patient outcomes (e.g., adherence, QOL) will be assessed in the included studies. Details on how outcome variables will be analyzed or summarized should be included.

The impact of interventions on patient outcomes (e.g., adherence, QOL) will not be evaluated in the included studies. It seems that this may not be the intended purpose of a scoping review. We believe that we have addressed this in the text.

The manuscript is generally well-written, but certain sections are repetitive and could be streamlined for clarity. For example, the age range of participants is mentioned multiple times. Additionally, some terms should be revised to align with person-first language (e.g., “schizophrenic patients” should be changed to “patients with schizophrenia”).

The changes were made following your instructions.

Several linguistic and conceptual inconsistencies should also be addressed:

• .. conditions are amongst the biggest public health issues, triggering both severe human .. could be streamlined to "one of the most significant public health issues"

• "The success of the nurse-led psychosocial therapies created and delivered by of nurse-led psychoeducational therapies created and delivered by advanced psychiatric practice nurses." A simpler version could be: "The success of nurse-led psychoeducational therapies delivered by advanced psychiatric practice nurses...”

• One of the research questions is: “What are the characteristics of nurse-led intervention programs? The characteristics to be searched for are the name of the program”. It should be “Extracted” instead of “searched”.

• The headline of a section is “review question”, but it is “research question” throughout the text

• “Qualitative studies will also be considered that focus on qualitative data including, but not limited to, designs such as phenomenology, grounded theory, ethnography, qualitative description, action research and feminist research.” However, these are technically not designs, but research methods/methodologies and paradigms.

• They switch between calling it papers/studies throughout the ms

All your suggestions have been taken into consideration, and we have amended the manuscript accordingly.

Several methodological aspects require further clarification:

• The ms states that,”The strengths of this review are the use of the PRISMA methodology for selection and the synthesis of the articles and the use of three different databases (Scopus, PubMed and Cinahl), primarily carried out by people with experiences in the sector”. It is not clear what it is that are primarily carried out by people with experiences in the sector.

Thank you for the suggestion. We added a clearer sentence in the manuscript.

• It is not clear why “Standard interventions and/or consultations provided by mental health services will be excluded”?

Because the scoping review included only studies on nurse-led psychoeducative interventions that are not considered standard interventions, nor as consultations.

• While the ms states that data will be publicly disseminated, it does not specify the repository or how the data will be made accessible. Please specify the repository where the data will be stored (e.g., OSF or Dryad) and outline measures for validating extracted data to ensure reliability.

We specified the repository where the data will be stored (OSF). It is important to note that scoping reviews do not summarize findings from included sources of evidence, as this is more appropriate in the context of a systematic review.

• The ms does not include a preliminary PRISMA flow diagram, which is a key component of scoping review protocols. A diagram outlining the planned study selection process should be added

We added the PRISMA flow diagram as a supplementary file.

While the conclusions appropriately address the knowledge gap in nurse-led psychoeducational interventions, certain claims are made prematurely without supporting evidence. Specifically, the assertion that these interventions improve patient outcomes and treatment adherence is made without data to substantiate it. Additionally, the manuscript overgeneralizes the expected impact of the review on clinical practice. The decision to exclude grey literature may result in the omission of relevant studies, particularly in a field where nurse-led interventions may not be widely published in peer-reviewed journals. The potential limitations of excluding grey literature should be acknowledged and discussed.

We added a section on strengths and limitations, where we discussed the reason for the exclusion of grey literature.

The ms covers an important and timely topic, but several methodological and conceptual issues need to be addressed before it can be recommended for publication. The authors should:

1. Clarify the rationale for conducting this review and ensure alignment between the research questions and study objectives.

2. Provide more detail on inclusion/exclusion criteria, study selection, and data extraction.

3. Address inconsistencies and redundancies in terminology and writing.

4. Strengthen the methodological framework by specifying data validation methods and including a PRISMA flow diagram.

5. Moderate claims about the effectiveness of interventions and discuss the limitations of excluding grey literature.

Reviewer #2: This is the protocol for a scoping review that will be conducted on studies implementing nurse-led psychoeducational interventions for patients and families of patients with psychotic spectrum disorders. The protocol is of interest; below are some brief comments:

The manuscript requires revision by a native English speaker. Examples of corrections: “Primary and secondary studies pertaining to both quantitative and qualitative paradigm.” This sentence appears incomplete; “Adults suffering from s schizophrenia”; “and or” should likely be “and/or”; “they do not appear to be present to date similar studies in the literature” should be rephrased; “When appropriate, authors of papers will be contacted to request missing” is an incomplete sentence; “we aspect” should probably be “expect.”

An attempt was made to reformulate the previously mentioned sentences, with the objective of enhancing the quality of the English language.

This is a protocol for a study that has not yet been conducted, so ensure that sentences are in the correct tense (e.g., “We carried out this scoping review” should be “We will carry out”).

We rephrased the sentence.

Points 4 and 5 under Strengths and Limitations seem to imply that the study has already been conducted.

We rephrased these sections.

The in-text reference numbers appear after the period of the sentence they refer to. Ensure this is the correct format.

It is the correct form.

“The success of the nurse-led psychosocial therapies created and delivered by of nurse-led psychoeducational therapies created and delivered by advanced psychiatric practice nurses for patients who were recently identified and referred to psychiatric care (especially for patients suffering from schizophrenia or other psychotic disorders), however, has been demonstrated by a relatively small number of controlled trials and, consequently, limited clinical data.” This sentence is hard to read and contains errors.

We reformulated the sentences. Thank you for the suggestion.

In the subquestions of the review questions, I would add patient diagnoses to point 3.

We inserted it in point 3 of the review questions.

Since it is already March 2025, I would update the target dates for the various steps of the review.

We updated the target dates for the various steps of the review. Now it will be September 2025.

To ensure the validity of results, more information regarding the data extraction tool developed by reviewers is necessary, particularly concerning its validation if it is an automated tool for parts of the process not supervised by humans.

We inserted more information regarding the data extraction tool developed by reviewers, as suggested.

The time frame on which the research will focus must be specified, as per PRISMA guidelines. Up to which month in 2025 will inclusion criteria for the search apply?

Please note that the sentences concerning the adherence of our scoping review to the PRISMA guidelines have been incorporated in the text. The completion of the screening phase was scheduled for July 2025._

In the supplementary materials regarding search strings, at least one string lists a year limit up to 2023.

We corrected them.

7. PLOS authors have the option to publish the peer review history of their article (what does this mean?). If published, this will include your full peer review and any attached files.

Do you want your identity to be public for this peer review? For information about this choice, including consent withdrawal, please see our Privacy Policy.

Reviewer #1: Yes: lars clemmensen

Reviewer #2: No

---

## [Decision Letter · Decision Letter 1]

Nurse-led Psychoeducational Interventions in Patients Suffering from Schizophrenia or Other Psychotic Disorders and their families: a Scoping Review Protocol.

PONE-D-24-58421R1

Dear Dr. gambaro,

We’re pleased to inform you that your manuscript has been judged scientifically suitable for publication and will be formally accepted for publication once it meets all outstanding technical requirements.

Kind regards,

Eleni Petkari

Academic Editor

PLOS ONE

Additional Editor Comments (optional):

Reviewers' comments:

Reviewer's Responses to Questions

**Comments to the Author**

1. Does the manuscript provide a valid rationale for the proposed study, with clearly identified and justified research questions?

Reviewer #1: Yes

2. Is the protocol technically sound and planned in a manner that will lead to a meaningful outcome and allow testing the stated hypotheses?

Reviewer #1: Yes

3. Is the methodology feasible and described in sufficient detail to allow the work to be replicable?

Reviewer #1: Yes

4. Have the authors described where all data underlying the findings will be made available when the study is complete?

Reviewer #1: Yes

5. Is the manuscript presented in an intelligible fashion and written in standard English?

Reviewer #1: Yes

You may also provide optional suggestions and comments to authors that they might find helpful in planning their study.

Reviewer #1: The authors have satisfactorily responded to my concerns, And I can now recommend the ms for publication

**Do you want your identity to be public for this peer review?** For information about this choice, including consent withdrawal, please see our Privacy Policy

Reviewer #1: **Yes: ** Lars Clemmensen

---

## [Editor Report · Acceptance letter]

PONE-D-24-58421R1

PLOS ONE

Dear Dr. gambaro,

I'm pleased to inform you that your manuscript has been deemed suitable for publication in PLOS ONE. Congratulations! Your manuscript is now being handed over to our production team.

Kind regards,

on behalf of

Dr. Eleni Petkari

Academic Editor

PLOS ONE